# Learning-based Motion Planning in Dynamic Environments Using GNNs and Temporal Encoding

**Ruipeng Zhang**
UCSD

**Chenning Yu**
UCSD

**Jingkai Chen**
MIT

**Chuchu Fan**
MIT

**Sicun Gao**
UCSD

## Abstract

Learning-based methods have shown promising performance for accelerating motion planning, but mostly in the setting of static environments. For the more challenging problem of planning in dynamic environments, such as multi-arm assembly tasks and human-robot interaction, motion planners need to consider the trajectories of the dynamic obstacles and reason about temporal-spatial interactions in very large state spaces. We propose a GNN-based approach that uses temporal encoding and imitation learning with data aggregation for learning both the embeddings and the edge prioritization policies. Experiments show that the proposed methods can significantly accelerate online planning over state-of-the-art complete dynamic planning algorithms. The learned models can often reduce costly collision checking operations by more than 1000x, and thus accelerating planning by up to 95%, while achieving high success rates on hard instances as well.

## 1 Introduction

Motion planning for manipulation has been a longstanding challenge in robotics [2, 12]. Learning-based approaches can exploit patterns in the configuration space to accelerate planning with promising performance [31, 6, 4]. Existing learning-based approaches typically combine reinforcement learning (RL) and imitation learning (IL) to learn policies for sampling or ranking the options at each step of the planning process [36, 47, 5]. Graph Neural Networks (GNNs) are a popular choice of representation for motion planning problems, because of their capability to capture geometric information and are invariant to the permutations of the sampled graph [21, 24, 25, 46].

Motion planning in dynamic environments, such as for multi-arm assembly and human-robot interaction, is significantly more challenging than in static environments. Dynamic obstacles produce trajectories in the temporal-spatial space, so the motion planner needs to consider global geometric constraints in the configuration space at each time step (Figure 1). This dynamic nature of the environment generates the much larger space of sequences of graphs for sampling and learning, and it is also very sensitive to the changes in one single dimension: time. A small change in the timing of the ego-robot or the obstacles in two spatially similar patterns may result in completely different planning problems. For instance, the dynamic obstacle may create a small time window for the ego-robot to pass through, and if that window is missed, then the topology of configuration space can completely change. Consequently, we need to design special architectures that can not only encode the graph structures well, but also infer temporal information robustly. Indeed, complete search algorithms for dynamic motion planning, such as the leading method of Safe Interval Path Planning (SIPP) and its variations [34, 26, 11, 30], focus on reasoning about the temporal intervals that are safe for the ego-robot. These complete algorithms typically require significantly more computation and collision checking operations compared to the static setting. As it is proved in [37], the computational complexity of planning with the moving obstacles is NP-hard even when the ego-robot has only a small and fixed number of degrees of freedom of movement.

36th Conference on Neural Information Processing Systems (NeurIPS 2022).

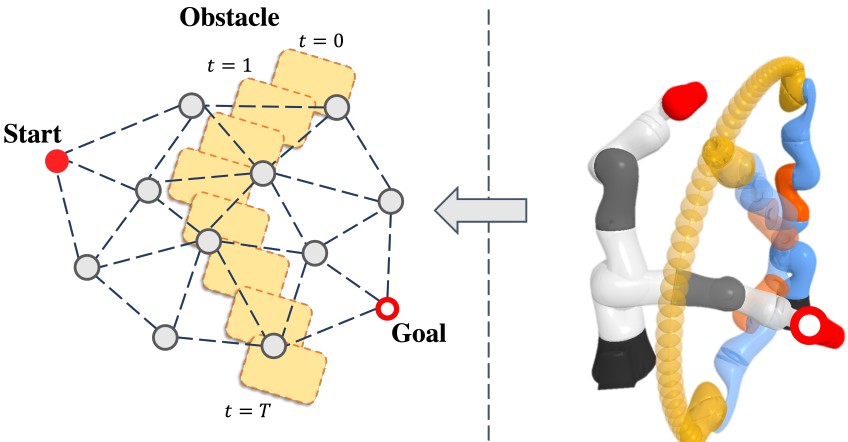

Figure 1: Left: A sampled graph from the configuration space. A dynamic obstacle, colored in yellow, moves over time from $t = 0$ to $t = T$. The goal of our approach is to search for a path on the graph connecting the start to the goal, without collision with the obstacle at any timestep. Right: A successful plan where the ego-robot (grey arm) avoids collision with the dynamic obstacle (blue arm) and reaches the goal.

We propose a novel Graph Neural Network (GNN) architecture and the corresponding training algorithms for motion planning in dynamic environments. We follow the framework of sampling-based motion planning [20, 19], where path planning is performed on random graphs sampled from the configuration space. The GNN takes in the following inputs: the sampled graph in the configuration space, the obstacle's trajectory in the workspace, and the current state of the ego-robot. The output is a vector of priority values on the candidate edges at the current state of the ego-robot. The encoding is performed in two stages. In the first stage, we encode the graph structure using attention mechanisms [43], and also design a temporal encoding approach for the obstacle trajectories. The temporal encoding uses the idea of positional encoding from the Transformer and NeRF [43, 28], which encourages the neural network to capture temporal patterns from high-frequency input signals. In the second stage of encoding, we incorporate the ego-robot's current vertex in the configuration space, the local graph structure, and the current time-shifted trajectories of the obstacles. This two-stage structure extends the previous use of GNNs in static environments [24, 25, 46] and it is important for making high-quality predictions on the priority values. The entire GNN of both stages will be trained simultaneously in an end-to-end fashion. Due to the complexity of the GNN architecture, we observe that RL-based approaches can hardly train generalizable models based on the architecture, and using imitation learning with data aggregation (DAgger) is the key to good performance. We utilize SIPP as the expert, first perform behavior cloning as warm-up, and then allow the ego-robot to self-explore and learn from the expert following the DAgger approach [38, 13].

We evaluate the proposed approach in various challenging dynamic motion planning environments ranging from 2-DoF to 7-DoF KUKA arms. Experiments show that our method can significantly reduce collision checking, often by more than 1000x compared to the complete algorithms, which leads to reducing the online computation time by over 95%. The proposed methods also achieve high success rates in hard instances and consistently outperform other learning and heuristic baselines.

## 2   Related Work

**Motion Planning in Dynamic Environments.** Planning in dynamic environment is fundamentally difficult [37]. Safe Interval Path Planning (SIPP) [34] and its variations [11, 30, 26] are the leading framework for motion planning in dynamic environments. The method significantly reduces the temporal-spatial search space by grouping safe configurations over a period of time *safe intervals*, [34]. Based on this data structure, it can find the optimal paths on a configuration graph. Further improvements on SIPP include leveraging state dominance [11] and achieving anytime search [30]. Recently, [26] shows SIPP is also powerful in planning the paths of high-dimensional manipulators on configuration roadmaps [20] with the presence of dynamic obstacles, which is the problem of

interests in this paper. Other methods have been proposed for path planning in dynamic environments for mobile robots specifically [14, 33, 7]. Such problems are typically more efficiently solvable because of the low-dimensional configuration space and the use of more conservative planning.

**Learning-based Motion Planning.** Learning-based approaches typically consider motion planning as a sequential decision-making problem that can be tackled with reinforcement learning or imitation learning. With model-based reinforcement learning, DDPG-MP [18] integrates the known dynamic of robots and trains a policy network. [40] improves obstacle encoding with the position and normal vectors. OracleNet [1] learns via oracle imitation and encodes the trajectory history by an LSTM [15]. Other than greedily predicting nodes that sequentially form a trajectory, various approaches have been designed to first learn to sample vertices, and then apply search algorithms on the sampled graphs [16]. [17, 36, 47] learn sampling distributions to generate critical vertices from configuration space. [27, 5, 23] design neural networks that can better handle structured inputs. Learning-based approaches have also been proposed to improve collision detection [6, 4, 46] and for exploration of edges on fixed graphs [41, 23]. For non-static environments, most works focus on multi-agent scenarios [32, 13] that do not involve non-cooperative dynamic obstacles.

**Graph Neural Networks for Motion Planning.** Graph neural networks are permutationally invariant to node ordering, which becomes a natural choice for learning patterns on graphs. For motion planning, [21] utilizes GNN to identify critical samples. [24, 25, 8] predicts the action for each agent in the grid-world environment using graph neural networks. [42, 22] learns the control policy for each robot in a large-scale swarm. [48] learns the submodular action selection for continuous space. [46] learns a GNN-based heuristic function for search in static environment with completeness guarantee.

## 3 Preliminaries

**Sampling-based Motion Planning with Dynamic Obstacles.** We focus on the sampling-based motion planning, in which a random graph is formed over samples from the *configuration space* $C \subseteq \mathbb{R}^n$ where $n$ is the number of degree-of-freedom for the ego-robot. The sampled vertex set $V$ always contains the start vertex $v_s$ and goal vertex $v_g$. The edges in the graph $G = \langle V, E \rangle$ are determined by r-disc or k-nearest-neighbor (k-NN) rules [10, 45]. We assume global knowledge of the trajectories of the dynamic obstacles. We represent the trajectories of dynamic obstacles in the *workspace* as the vector of all the joint positions in the time window of length $T > 0$. The goal of the motion planning problem is to find a path from $v_s$ to $v_g$ in the sampled graph that is free of collision with the dynamic obstacles at all time steps in $[0, T]$.

**Graph Neural Networks (GNNs).** GNNs learn representations over of vertices and edges on graphs by message passing. With MLP networks $f$ and $g$, GNN encodes the representation $h_i^{(k+1)}$ of vertex $v_i$ after $k$ aggregation steps defined as

$$h_i^{(k+1)} = g(h_i^{(k)}, \oplus(\{f(h_i^{(k)}, h_j^{(k)}) \mid (v_i, v_j) \in E\})) \tag{1}$$

where $h_i^{(1)} = x_i$ can be some arbitrary vector of initial data for the vertex $v_i$. $\oplus$ is typically some permutation-invariant aggregation function on sets, such as mean, max, or sum. We use the attention mechanism to encode the obstacle features. In the general form of the attention mechanism, there are $n$ keys, each with dimension $d_k$: $K \in \mathbb{R}^{n \times d_k}$, each key has a value $V \in \mathbb{R}^{n \times d_v}$. Given $m$ query vectors $Q \in \mathbb{R}^{m \times d_k}$, we use a typical attention function $\mathbf{Att}(K, Q, V)$ for each query as $\mathbf{Att}(K, Q, V) = \mathrm{softmax}(QK^T/\sqrt{d_k})V$ [43].

**Imitation Learning.** Imitation learning aims to provide guidance to train policy without explicitly designing reward functions. Given a distribution of the oracle actions $\pi_{oracle}$, it tries to learn a new policy distribution $\pi$ that minimize the deviation from the oracle, i.e. $\pi^* = \mathrm{argmin}_\pi D(\pi, \pi_{oracle})$, where $D$ is the difference function between the two distributions, which can be represented as $p$-norm or $f$-divergence. We use imitation learning to train our GNN models from the demonstration of the oracle planner. Specifically, in the task of sampling-based motion planning, the oracle predicts the priority values of subsequent edges and prioritize the one given by the oracle. However, imitation learning based on behavior cloning often suffers from distribution drift, which can be mitigated by imitation with data aggregation (DAgger) [38]. With DAgger, the learner can actively query the oracle on states that are not typically provided in the expert trajectories to robustify the learned policy.

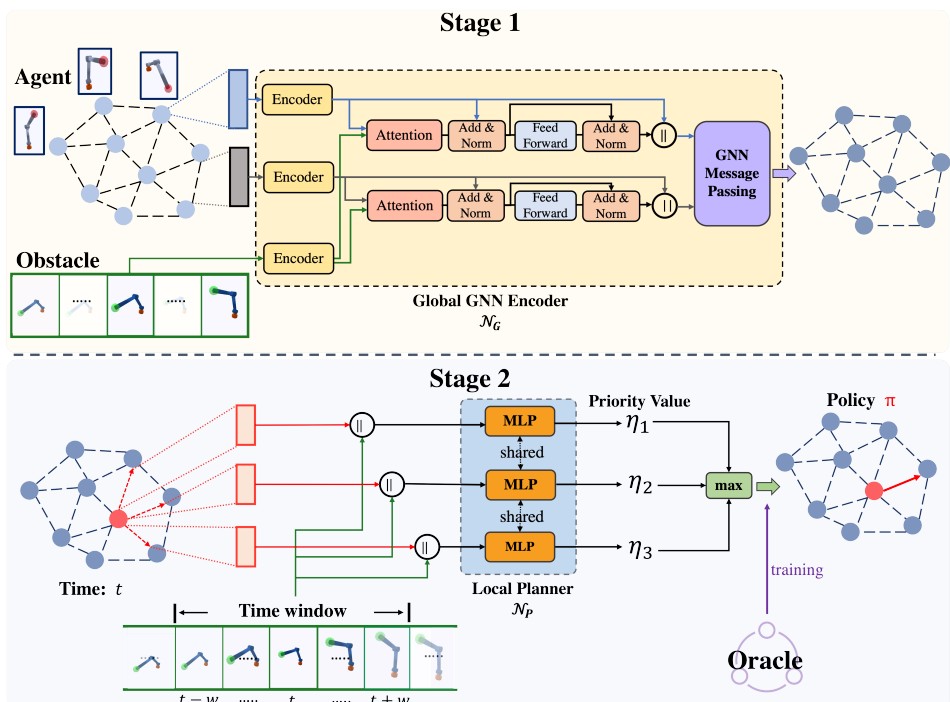

Figure 2: The overall two-stage architecture of the proposed **GNN-TE**. In Stage 1, we encode global information of the ego-arm and the obstacles, using attention mechanisms, and output the encoding of each edge. In Stage 2, the local planner will take in the output from Stage 1 along with the obstacle encoding within the relevant time window, to predict the priority value of each outgoing edge. The planner will propose the edge with the highest priority value to take as the output policy.

## 4 GNN-TE: Dynamic Motion Planning with GNNs and Temporal Encoding

### 4.1 Overall Architecture

We design the dynamic motion planning network **GNN-TE** to capture the spatial-temporal nature of dynamic motion planning. The forward pass in the network consists of two stages. The first stage is the global GNN encoder $\mathcal{N}_G$ that encodes the global information of the ego-robot and obstacles in the environment. The second stage is the local planner $\mathcal{N}_p$ that assigns priority on edges, utilizing the encoding output of the first stage. Fig 2 shows the overall two-stage architecture.

**First-stage global GNN encoder** $\mathcal{N}_G$. The GNN encoder $\mathcal{N}_G$ takes in a sampled random geometric graph $G = \langle V, E \rangle$, $V = \{v_s, v_g, v\}$. For an $n$-dimensional configuration space, each vertex $v_i \in \mathbb{R}^{n+1}$ contains an $n$-dimensional configuration component and a 1-dimensional one-hot label indicating if it is the special goal vertex.

The vertices and the edges are first encoded into a latent space with $x \in \mathbb{R}^{|V| \times d_h}, y \in \mathbb{R}^{|E| \times d_h}$, where $d_h$ is the size of the encoding. Specifically, to get the feature $x_i$ for the $i$-th node $v_i \in V$, we use $x_i = g_x(v_i, v_g, v_i - v_g, ||v_i - v_g||_2^2)$. To get the feature $y_l$ for the $l$-th edge $e_l : \langle v_i, v_j \rangle \in E$, we use $y_l = g_y(v_i, v_j, v_j - v_i)$. The $g_x$ and $g_y$ are two different two-layer MLPs. The L2 distance to the goal $||v - v_g||_2^2$ serves as the heuristic information for $\mathcal{N}_G$.

The dynamic obstacles $O$ form barriers on the top of the graph $G$, and we incorporate their trajectories to $\mathcal{N}_G$ to leverage the global environment and assist the planning of $\mathcal{N}_P$. Additionally, to inform the networks about the relative time over the horizon, we incorporate *temporal encoding* with the obstacles. Given the obstacle position $O_t$ at time step $t$ and a two-layer MLP $g_o$, the obstacle is encoded as $\mathcal{O}_t = g_o(O_t) + TE(t)$, which adds $g_o(O_t)$ and $TE(t)$ element-wisely. $TE(t)$ is the temporal encoding at time step $t$, which we will discuss at Section 4.2. With the sequential property of the trajectory, we use the attention mechanism to model the temporal-spatial interactions between

the ego-arm and the obstacles. Concretely, the obstacles are encoded into the vertex and edge of $G$ as:

$$x = x + \mathbf{Att}(f_{K_x^{(i)}}(\mathcal{O}), f_{Q_x^{(i)}}(x), f_{V_x^{(i)}}(\mathcal{O})) \tag{2}$$

$$y = y + \mathbf{Att}(f_{K_y^{(i)}}(\mathcal{O}), f_{Q_y^{(i)}}(y), f_{V_y^{(i)}}(\mathcal{O})) \tag{3}$$

Taking the vertex and edge encoding $x, y$, the GNN $\mathcal{N}_G$ aggregates the local information for each vertex and edge from the neighbors with the following operation with 2 two-layer MLPs $f_x$ and $f_y$:

$$x_i = \max\left(x_i, \max\{f_x(x_j - x_i, x_j, x_i, y_l) \mid e_l : \langle v_i, v_j \rangle \in E\}\right), \forall v_i \in V$$
$$y_l = \max(y_l, f_y(x_j - x_i, x_j, x_i)), \forall e_l : \langle v_i, v_j \rangle \in E \tag{4}$$

Note that here we use $\max$ as the aggregation operator to gather the local geometric information, due to its empirical robustness to achieve the order invariance [35]. The edge information is also incorporated into the vertex by adding $y_l$ as the input to $f_x$. Also, because Equation 4 is a homogeneous function that updates on the $x$ and $y$ in a self-iterating way, we can update without introducing redundant layers over multiple loops. After several iterations, the first-stage $\mathcal{N}_G$ outputs the encoding of each vertex $x_i$ and edge $y_l$.

**Second-stage local planner** $\mathcal{N}_P$. After $\mathcal{N}_G$ encodes the information of the configuration graph and obstacles, the second-stage local planner $\mathcal{N}_P$ utilizes the encoding and performs motion planning. Specifically, when arriving at a vertex $v_i$ at time $t_i$, $\mathcal{N}_P$ predicts the priority value $\eta_{e_i}$ of all the connected edges $e_i \in E_i$ with the expression $\eta_{e_i} = f_p(y_{e_i}, \mathcal{O}_{t_i-w}, \mathcal{O}_{t_i-w+1}, ..., \mathcal{O}_{t_i+w-1}, \mathcal{O}_{t_i+w})$, where $f_p$ is an MLP. Note that in addition to the encoding of the connected edges, we also input the local obstacle encoding within a time window $w$ of the current arrival time $t_i$. This provides local information for $\mathcal{N}_P$ to plan towards the goal vertex, while considering the barriers of dynamic obstacles to avoid collisions. At inference time, we use $\mathcal{N}_P$ to choose the edge with the highest priority value while keeping track of the current time.

## 4.2 Temporal Encoding

Positional encoding is a crucial design in the Transformer architecture [44] for making use of the order of the sequence. Dynamic motion planning requires the models to infer the relative position of obstacles and how they interact with the ego-arm at each time step. So along with the positions of the obstacles in the workspace, we add temporal encoding $TE(t) \in \mathbb{R}^{d_{TE}}$ at each time step $t \in [0, \cdots, T]$, where its $2k$-th and $2k+1$-th dimensions are computed as

$$TE(t, 2k) = \sin(\omega^{-2k/d_{TE}}t) \text{ and } TE(t, 2k+1) = \cos(\omega^{-2k/d_{TE}}t) \tag{5}$$

where $\omega \in \mathbb{Z}^+$ is a fixed frequency. We select the temporal encoding to have the same dimension as the obstacle input, and add them as the obstacle encoding before inputting into the networks $\mathcal{N}_G$ and $\mathcal{N}_P$. We illustrate the overall encoding procedure on the left of Figure 3.

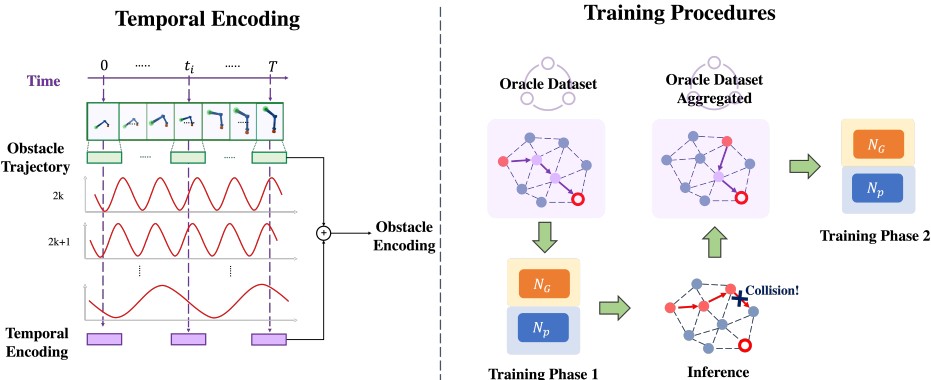

Figure 3: Left: Temporal encoding is incorporated when representing the dynamic obstacle sequence. Right: Training procedures with DAgger. The proposed GNN is first trained to imitate an optimal oracle, then improves itself by self-exploring the data with feedback from the oracle.

### 4.3 Training and Inference Procedures

In each training problem, along with the dynamic obstacles $\mathcal{O}$, start vertex $v_s$ and goal vertex $v_g$, we sample a k-NN graph $G = \langle V, E \rangle, V = \{v_s, v_g, v\}$, where $v$ is the vertices sampled from the configuration space of the ego-arm. In the first stage, we use the global GNN encoder $\mathcal{N}_G$ to encode the graph and dynamic obstacles, and the local planner $\mathcal{N}_p$ in the second stage uses the encoding as the input to predict the priority value $\eta$ of the subsequent edges.

**Imitation from SIPP with Data Aggregation.** We train our two-stage network $\mathcal{N}_G$ and $\mathcal{N}_p$ in an end-to-end manner by imitating an oracle. Specifically, we use Safe Interval Path Planning (SIPP) [34] to compute the shortest non-collision motion path and use it as the oracle.

In the first stage, $\mathcal{N}_G$ will process the graph and the obstacle trajectories, then output the encoded features of each vertex and edge. Then in the second stage, we train the networks to imitate the oracle SIPP along the optimal path. Concretely, starting from the $v_s$, SIPP provides the optimal path $\pi^* = \{(v_i^*, t_i^*)\}_{i \in [0,n]}$ with the vertex $v_i^*$ and the corresponding arrival time $t_i^*$. When arriving at the vertex $v_i^*$ at time $t_i^*$, the local planner $\mathcal{N}_p$ will take in the edge feature $y_{e_i}$ along with the obstacle encoding in the time window $[t_{i-w}^*, t_{i+w}^*]$ to predict the priority value of all the subsequent edges $E_i$. Then it prioritizes the next edge on the optimal path $e_i^* : \langle (v_i^*, t_i^*), (v_{i+1}^*, t_{i+1}^*) \rangle$ among $E_i$. We maximize the priority value $\eta_{e_i^*}$ of $e_i^*$ over all other $e_i \in E_i \setminus \{e_i^*\}$ with the standard cross entropy loss $L_i = -\log(\exp(\eta_{e_i^*})/(\Sigma_{e_i \in E_i} \exp(\eta_{e_i})))$.

Since SIPP only provides the guidance on the optimal path, when the planned path given by $\mathcal{N}_p$ deviates from the optimal path, our network cannot sufficiently imitate the oracle. To this end, we use DAgger [38] to encourage the network to learn from these sub-optimal paths. We first train our network for $k$ iterations with pure demonstrations from SIPP. Then we explore a path $\pi^k$ on the graph using the priority value predicted by the current network, which may not reach the goal vertex $v_g$ nor be optimal. We randomly stop at the vertex $v_i^k$ at time $t_i^k$ and query the oracle. SIPP treats $v_i^k$ and $t_i^k$ as the start vertex and the start time respectively, along with the obstacles trajectory starting at $t_i^k$, calculates the optimal path. The new demonstrations are aggregated to the previous dataset to keep training the network. The training procedures are showed on the right of Figure 3.

**Inference with Time Tracking.** Given a graph $G = \langle V, E \rangle, V = \{v_s, v_g, v\}$, with the trajectories of obstacles $\mathcal{O}$, $\mathcal{N}_G$ will first encode the graph and obstacles. Next, $\mathcal{N}_P$ executes motion planning by predicting the priority values $\eta = \mathcal{N}_P(V, E_i, \mathcal{O}, \mathcal{N}_G, t_i)$ when arriving at vertex $v_i$ at time $t_i$, and follow the edge with the maximum one, i.e. $e_{\pi_i} = \arg\max_{e_i \in E_i} \eta_{e_i}$. After the edge $e_{\pi_i}$ is proposed by the network, we check the collision on $e_{\pi_i}$ on which the ego-arm starts moving at $t_i$. If there is no collision, we add $(e_{\pi_i}, t_i)$ into the current path $\pi$. Otherwise, we query $\mathcal{N}_P$ for another edge with the next greater priority value. The planning will end if we succeed in finding a path $\pi$ from $v_s$ to $v_g$ or fail when stopping at a vertex with all the connected edges with collisions. Optionally, when the latter one happens, we can backtrack to the last step and query for the top-k greatest priorities in turn. Further discussions are covered in the experiments section.

## 5 Experiments

**Experiment Setup.** We evaluate the proposed methods on various multi-arm assembly tasks where we manipulate the ego-arm and avoid collisions with other obstacles, including moving arms and static obstacles. Specifically, the environments cover arms of 2 to 7 Degree of Freedom and 1 to 3 moving obstacle arms, including (i) 2Arms: 1 obstacle arm with 2 DoF. (ii) Kuka-4DoF: 1 obstacle arm with 4 DoF. (iii) Kuka-5DoF: 1 obstacle arm with 5 DoF. (iv) Kuka-7DoF: 1 obstacle arm with 7 DoF. (v) 3Arms: 2 obstacle arms with 2 DoF. (vi) Kuka3Arms: 2 obstacle arms with 7 DoF. Without loss of generality, we assume all the robot arms in an environment are of the same DoF and move at the same and constant speed. The static obstacles are represented as cuboids in the experiments.

**Baselines.** We compare our method **GNN-TE** with the lazy sampling-based dynamic motion planning method **Dijkstra-Heuristic** (**Dijkstra-H**). It prioritizes the edges based on the shortest distance to the goal on the configuration graph at each time step. We also compare our method with the oracle **SIPP** [34], which is a search algorithm that generates optimal paths using safe intervals.

**Datasets.** We randomly generate 2000 problems for training and 1000 problems for testing. In each problem, we randomly generate fixed-length trajectories of the moving obstacle arms and sample

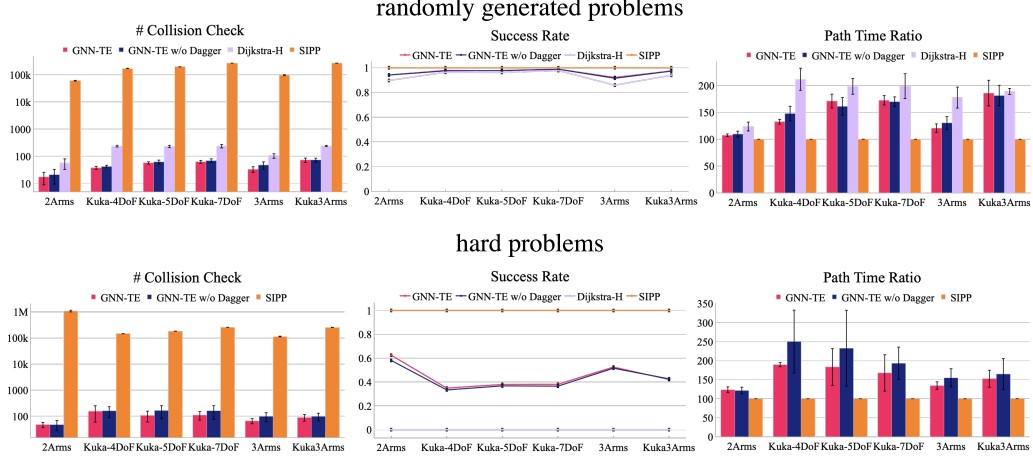

Figure 4: Overall Performance of baselines on collision checking, success rate, and path time ratio. Our approach significantly reduces collision checking more than 1000x compared to the complete algorithm SIPP, improves the overall planning efficiency, and achieves high success rates.

1000 vertices in the configuration space of the ego-arm that are collision-free with the static obstacles, then provide them to the oracle to generate demonstration trajectories. In order to further investigate the performance of algorithms in challenging environments, we also generate 1000 hard problems, where **Dijkstra-H** fails to find feasible paths.

**Training and Testing Details.** We first train the **GNN-TE** on all the training problems for 200 epochs. Afterward, we generate 1000 new training data with DAgger, and trained for another 100 epochs. We test all the algorithms on the provided graphs. We randomly split the test problems into 5 groups, and calculate the performance variance.

**Evaluation Metrics.** We measure the average number of **collision checking** of the common success cases among all the algorithms to evaluate the effectiveness in avoiding obstacles. Collision checking is the more expensive computation in motion planning, and reducing it can lead to a significant acceleration of online computation. We also report the average **success rate** over all the testing problems for all the algorithms. A trajectory is successful, only if the ego-arm does not collide with any obstacles, and eventually reaches the goal given a limited time horizon. We also measure the optimality of the methods by comparing the **path time ratio** to the optimal paths found by SIPP on each success case. Note that the path time metric only measures the cost of the planned path, and does not include the online computation time needed.

## 5.1 Overall Performance

Figure 4 shows the overall performance of the algorithms in various environments, including all the baselines and **GNN-TE** without DAgger, i.e., pure imitation learning from the oracle.

In all the environments, **SIPP** gives the optimal complete non-collision path, but it suffers from the excessive amount of collision checking. **GNN-TE** significantly reduces the collision checking by more than 1000x, which corresponds to reducing online computation time by over 95%. At the same time, the methods have high success rates and a better path time ratio compared to simpler heuristics.

Fig. 5 shows the performance snapshots of the algorithms on a 2Arms and a Kuka-7DoF test case. In both cases, our method successes in planning a near-optimal path compared to the oracle **SIPP** whereas **Dijkstra-H** fails.

**Performance on randomly generated test problems.** As shown in Fig. 4, our methods significantly reduces the collision checking by over 1000x compared to **SIPP**, i.e., 60k, 97k, 171k, 197k, 269k, 269k to 17.24, 33.08, 37.36, 56.89, 61.99, 72.63 on 2Arms, 3Arms, Kuka-4DoF, Kuka-5DoF, Kuka-7DoF, Kuka3Arms respectively. Because **SIPP** needs to search in both space

and time dimensions, it suffers from a large number of collision checking. Our approach benefits from learning and achieves a much less amount of collision checking even in the dynamic setting while not compromising much of the success rate, in which case our method outperforms **Dijkstra-H** in all the environments with higher success rates. Note that as a crucial part of the training procedures, DAgger also improves performance in all environments.

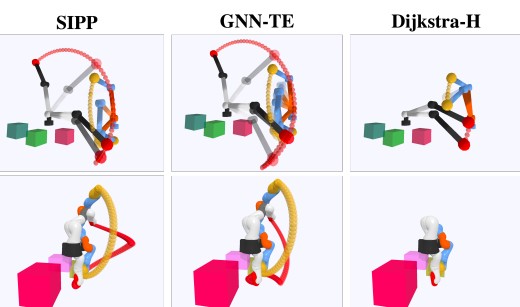

SIPP     GNN-TE     Dijkstra-H

**Performance on hard test problems.** On the hard test problems, **Dijkstra-H** fails to find feasible paths in the dynamic environment. Comparatively, our **GNN-TE** can successfully find solutions to these problems with considerable success rates and acceptable path time ratios. It is also worth noting that DAgger can better assist in improving the performance in the more challenging scenarios compared to the randomly generated problems.

Figure 5: Snapshots of the trajectories of the output path on 2 test cases. The environments are 2Arms (first row) and Kuka-7DoF (second row). The ego-arm is black and white with a red end-effector. The obstacle arm is blue and orange with a yellow end-effector. In both environments, **Dijkstra-H** fails to find a path, while our method can yield a near-optimal path compared to the oracle **SIPP**.

**Optional backtracking search.** In the Appendix C.3, we also report the result of **GNN-TE** and **Dijkstra-H** and them with backtracking **GNN-TE w. BT** and **Dijkstra-H w. BT** on 2Arms. On the same algorithm, the optional backtracking search will only result in a higher success rate while not affecting the path time ratio and collision checking on the common success cases. The result of **Dijkstra-H w. BT** shows that although it improves the success rate significantly but sacrificing a tremendous number of collision checking. Nevertheless, our method outperforms the heuristic method both with or without the backtracking search.

**Comparison with end-to-end RL.** We also compare our approach with RL approaches, including DQN [29] and PPO [39]. The neural network architectures for these two baselines are implemented with the same GNN architectures as ours. We observe that even in 2Arms, the average success rate of DQN on the training set is only around 54%, while PPO only has a success rate of around 20%. They fail to find plans in test cases. We provide more details in the Appendix C.4.

**Comparison with OracleNet-D.** We compare **GNN-TE** with a learning-based approach **OracleNet-D**, by modifing **OracleNet**[1] to the dynamic version. We observe that the performance of **OracleNet-D** falls behind **GNN-TE** largely on all the metrics both in random and hard problems. Details are provided in the Appendix C.5.

## 5.2 Ablation Studies

We perform ablation studies on 2Arms of the different encoding in our model, including global and local obstacle encoding and temporal encoding. The results are shown in Fig. 6.

**Global Obstacle Encoding.** In stage 1, we encode both the configuration graph and the global obstacle trajectories with the GNN-based global encoder. To investigate the effectiveness of leveraging global obstacle information, we conduct an experiment in which we input the sampled configurations in stage 1 and only introduce the obstacle information in stage 2. As we can observe in the figure that although on random problems there are slight degrades of collision checking and success rate, the performance of path time ratio, success rate, and collision checking improves greatly in the hard environment. This results from the model receiving the overall trajectories of the obstacles, and the encoding helps to reduce collision checking and improves success rate, especially on complicated problems.

**Local Obstacle Encoding.** We compare our model with the one omitting the local obstacle encoding in stage 2. We only input the temporal encoding corresponding to the arrival time to the local planner as the time indicator for planning. The result has shown that the local obstacle encoding within a time window directly helps the local planner perform better.

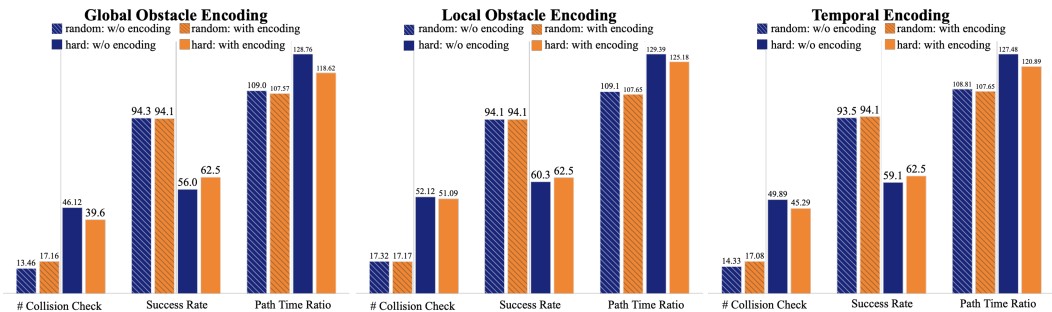

Figure 6: Ablation studies on 2Arms of (1) global obstacle encoding, (2) local obstacle encoding, and (3) temporal encoding. We have demonstrated that all these three components improve the effectiveness of the proposed approach. See Section 5.2 for more details.

**Temporal Encoding.** We analyze the importance of temporal encoding where we remove it from both two stages and only input the trajectory of obstacles in the models. The results also show that temporal information helps **GNN-TE** make use of the order of the trajectory sequence on both random and hard problems.

# 6 Discussion and Conclusion

We proposed a GNN-based neural architecture **GNN-TE** for motion planning in dynamic environments, where we formulate the spatial-temporal structure of dynamic planning problem with GNN and temporal encoding. We also use imitation learning with DAgger for learning both the embedding and edge prioritization policies. We evaluate the proposed approach in various environments, ranging from 2-DoF arms to 7-DoF KUKA arms. Experiments show that the proposed approach can reduce costly collision checking operations by more than 1000x and reduces online computation time by over 95%. Future steps in this direction can involve using ideas from the complete planning algorithms, such as incorporating safe intervals, to improve the success rate on hard instances, as well as more compact architectures for further reducing online computation.

# 7 Acknowledgement

This material is based on work supported by DARPA Contract No. FA8750-18-C-0092, AFOSR YIP FA9550-19-1-0041, NSF Career CCF 2047034, NSF CCF DASS 2217723, and Amazon Research Award. We appreciate the valuable feedback from Ya-Chien Chang, Milan Ganai, Chiaki Hirayama, Zhizhen Qin, Eric Yu, Hongzhan Yu, Yaoguang Zhai, and the anonymous reviewers.

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
