# Appendix A    Algorithms

---

**Algorithm 1:** Stage1: Global GNN Encoder $\mathcal{N}_G$

---

**Input:** obstacles $O$, start $v_s$, goal $v_g$, network $g_x, g_y, K_x^{(i)}, Q_x^{(i)}, V_x^{(i)}, K_y^{(i)}, Q_y^{(i)}, V_y^{(i)}$ for $i$-th embedding dimension.

Sample $n$ nodes $v_1, \cdots, v_n$ from configuration space of ego-arm robot

Initialize $G = \{V : \{v_s, v_g, v_1, \cdots, v_n\}, E : \text{k-NN}(V)\}$

Initialize encoding of vertices and edges

$$x_i = g_x(v_i, v_g, v_i - v_g, ||v_i - v_g||_2^2), \forall v_i \in V$$
$$y_l = g_y(v_i, v_j, v_j - v_i), \forall e_l : \langle v_i, v_j \rangle \in E$$

Initialize obstacle encoding $\mathcal{O}_t = g_o(O_t) + TE(t), \forall t \in [0, \cdots, T]$ using Eq. 5

Encode obstacles into vertices and edges using Eq. 3

Message Passing using Eq. 4

**return** Encoding of edges $\{y_l\}$

---

**Algorithm 2:** Stage2: Lobal Planner $\mathcal{N}_P$

---

**Input:** graph $G = \langle V, E \rangle$, encoding of edges $y_l$, obstacle encoding $\mathcal{O}$, time window $w$, global GNN encoder $\mathcal{N}_G$, local planner $\mathcal{N}_P$, goal-reaching constant $\delta$.

Initialize $i = 0, v_0 = v_s, t_0 = 0, \pi = (v_0, t_0), E_0 = \{e : \langle v_s, v_k \rangle \in E, \forall v_k \in V\}$

**repeat**

  $\eta = \mathcal{N}_P(V, E_i, \mathcal{O}, \mathcal{N}_G, t_i)$

  select $e_j = \arg\max_{e_l \in E_i} \eta_l$, and $e_j$ connects $\langle v_i, v_j \rangle$

  **if** $e_j$ is collision-free when start moving from $t_i$

    $t_{i+1} = t_i + \Delta(v_i, v_j)$ ;        // $\Delta(v_i, v_j)$ is the travel time from $v_i$ to $v_j$

    $\pi_{i+1} \leftarrow \pi_i \cup \{(v_j, t_{i+1})\}$

    $v_{i+1} \leftarrow v_j$

    $E_{i+1} = \{e : \langle v_{i+1}, v_k \rangle \in E, \forall v_k \in V\}$

    **if** $||v_{i+1} - v_g||_2^2 \leq \delta$

      **return** $\pi$

    $i \leftarrow i + 1$

  **else**

    $E_i = E_i \setminus e_j$

**until** $E_i = \emptyset$

**return** $\emptyset$

---

**Algorithm 3:** Dijkstra-H

---

**Input:** graph $G = \langle V, E \rangle$, start $v_s$, goal $v_g$, goal-reaching constant $\delta$.

Sample $n$ nodes $v_1, \cdots, v_n$ from configuration space of ego-arm robot.

Initialize $G = \{V : \{v_s, v_g, v_1, \cdots, v_n\}, E : \text{k-NN}(V)\}$

Calculate the shortest distance $d_{v_k}$ on the graph from $v_g$ to each node $v_k \in V$ using Dijkstra's algorithm.

Initialize $i = 0, v_0 = v_s, t_0 = 0, \pi = (v_0, t_0), E_0 = \{e : \langle v_s, v_k \rangle \in E, \forall v_k \in V\}$

**repeat**

  select $v_j = \arg\min_{\langle v_i, v_j \rangle \in E_i} d_{v_j}$

  **if** $\langle v_i, v_j \rangle$ is collision-free when start moving from $t_i$

    $t_{i+1} = t_i + \Delta(v_i, v_j)$ ;        // $\Delta(v_i, v_j)$ is the travel time from $v_i$ to $v_j$

    $\pi_{i+1} \leftarrow \pi_i \cup \{(v_j, t_{i+1})\}$

    $v_{i+1} \leftarrow v_j$

    $E_{i+1} = \{e : \langle v_{i+1}, v_k \rangle \in E, \forall v_k \in V\}$

    **if** $||v_{i+1} - v_g||_2^2 \leq \delta$

      **return** $\pi$

    $i \leftarrow i + 1$

  **else**

    $E_i = E_i \setminus e_j$

**until** $E_i = \emptyset$

**return** $\emptyset$

# Appendix B   Network Architecture Details

We provide the numbers of network parameters in Table 1. Please refer to 2 for the overall two-stage architecture of the proposed **GNN-TE**.

Table 1: Network Architecture Details

| Name | Model |
|------|-------|
| **Stage1 Global GNN Encoder** | |
| Node Encoder Net $g_x$ | MLP((config_size+1)*4,32),MLP(32,32) |
| Edge Encoder Net $g_y$ | MLP((config_size+1)*3,32),MLP(32,32) |
| Obstacle Encoder Net $g_o$ | MLP(obstacle_size,32), MLP(32,32) |
| Attention Net | Key Network $f_{K_{(\cdot)}}$: MLP(32,32) |
| | Query Network $f_{Q_{(\cdot)}}$: MLP(32,32) |
| | Value Network $f_{V_{(\cdot)}}$: MLP(32,32) |
| Feedforward Net | MLP(32,32),MLP(32,32) |
| Node Message Passing $f_x$ | MLP(32*4,32),MLP(32,32) |
| Edge Message Passing $f_y$ | MLP(32*3,32),MLP(32,32) |
| **Stage2 Local Planner** | |
| Planner Net $f_P$ | MLP(32+obstacle_size*window_size, 64),MLP(64,32), MLP(32,32), MLP(32,1) |

# Appendix C   Experiments

## C.1   Hyperparameters

We provide the hyperparameters in Table 2.

Table 2: Hyperparameters

| Hyperparameters | Values |
|-----------------|--------|
| $k$ for k-NN | 50 |
| Training Epoch before DAgger | 200 |
| Training Epoch for DAgger | 100 |
| Learning Rate | 1e-3 |
| Temporal Encoding Frequency $\omega$ | 10000 |
| $d_{TE}$ | 32 |
| Time Window $w$ | 2 |

## C.2   Overall Performance

We provide the detailed overall performance in Table 3, 4 and 5.

Table 3: Success Rate (%)

| | | 2Arms | Kuka-4DoF | Kuka-5DoF | Kuka-7DoF | 3Arms | Kuka3Arms |
|---|---|-------|-----------|-----------|-----------|-------|-----------|
| **SIPP** | random | 100±0.00 | 100±0.00 | 100±0.00 | 100±0.00 | 100±0.00 | 100±0.00 |
| | hard | 100±0.00 | 100±0.00 | 100±0.00 | 100±0.00 | 100±0.00 | 100±0.00 |
| **GNN-TE** | random | **94.1±0.02** | **97.8±0.01** | 97.6±0.00 | **98.8±0.01** | **92.1±0.01** | **97.4±0.01** |
| | hard | **62.5±0.02** | **34.9±0.00** | **37.9±0.15** | **38.1±0.12** | **52.5±0.03** | 42.1±0.08 |
| **GNN-TE w/o Dagger** | random | **94.1±0.01** | 97.5±0.01 | **97.7±0.00** | **98.8±0.01** | 91.5±0.01 | 97.3±0.01 |
| | hard | 58.1±0.03 | 33.3±0.00 | 36.8±0.14 | 36.5±0.11 | 51.6±0.05 | **42.6±0.08** |
| **Dijkstra-H** | random | 89.7±0.03 | 96.3±0.01 | 96.2±0.01 | 97.7±0.01 | 85.9±0.01 | 93.9±0.01 |
| | hard | 0.00±0.00 | 0.00±0.00 | 0.00±0.00 | 0.00±0.00 | 0.00±0.00 | 0.00±0.00 |

Table 4: Path Time Ratio

| | | 2Arms | Kuka-4DoF | Kuka-5DoF | Kuka-7DoF | 3Arms | Kuka3Arms |
|---|---|---|---|---|---|---|---|
| **SIPP** | random | 100±0.00 | 100±0.00 | 100±0.00 | 100±0.00 | 100±0.00 | 100±0.00 |
| | hard | 100±0.00 | 100±0.00 | 100±0.00 | 100±0.00 | 100±0.00 | 100±0.00 |
| **GNN-TE** | random | **107.55±2.33** | **132.71±4.46** | 171.39±12.94 | 172.83±8.77 | **120.76±7.89** | 186.18±23.96 |
| | hard | 123.31±7.71 | **189.54±5.41** | **183.33±48.26** | 167.65±47.7 | **134.54±9.76** | **152.52±22.31** |
| **GNN-TE w/o Dagger** | random | 109.67±5.46 | 148.1±13.59 | **161.40±16.50** | 170.05±9.11 | 130.48±11.74 | **181.74±18.98** |
| | hard | **121.41±8.85** | 250.26±82.5 | 232.67±99.79 | 193.12±42.4 | 154.85±23.81 | 164.67±40.65 |
| **Dijkstra-H** | random | 123.73±8.29 | 212.09±20.73 | 198.72±14.7 | 199.22±23.22 | 177.88±19.59 | 189.23±5.64 |
| | hard | / | / | / | / | / | / |

Table 5: Collision Checking

| | | 2Arms | Kuka-4DoF | Kuka-5DoF | Kuka-7DoF | 3Arms | Kuka3Arms |
|---|---|---|---|---|---|---|---|
| **SIPP** | random | 60440.21±1543.21 | 171336.68±2061.60 | 196567.99±1152.81 | 268602.98±780.07 | 96713.81±3945.07 | 269033.61±1159.78 |
| | hard | 1080768.34±81176.44 | 145280.09±1448.0 | 182696.61±1271.86 | 257783.45±742.83 | 114337.00±3560.95 | 255173.7±2099.46 |
| **GNN-TE** | random | **17.24±8.45** | **37.36±4.74** | **56.89±5.41** | **61.99±8.08** | **33.08±7.95** | **72.63±13.13** |
| | hard | **47.31±7.72** | **155.7±96.49** | **108.25±48.29** | **110.65±39.63** | **65.93±14.84** | **90.42±25.61** |
| **GNN-TE w/o Dagger** | random | 21.13±11.71 | 42.00±4.60 | 61.79±10.39 | 69.39±10.67 | 47.48±14.40 | 73.33±11.57 |
| | hard | 47.91±8.85 | 160.43±70.64 | 166.61±86.29 | 164.41±88.98 | 98.67±37.69 | 98.41±32.15 |
| **Dijkstra-H** | random | 56.21±23.56 | 236.50±17.60 | 229.35±21.66 | 236.95±34.93 | 103.74±19.57 | 237.93±12.83 |
| | hard | / | / | / | / | / | / |

## C.3 Optional Backtracking Search

We provide the results of **GNN-TE** and **Dijkstra-H** with backtracking (top-5) in 2Arms environment in 6. Our method outperforms the heuristic method with and without the backtracking search.

Table 6: The performance of backtracking search in the 2Arms environment

| | | SIPP | Dijkstra-H | GNN-TE | Dijkstra-H w. BT | GNN-TE w. BT |
|---|---|---|---|---|---|---|
| **Success Rate** | random | 100% | 89.70% | **94.10**% | 94.10% | **98.00%** |
| | hard | 100% | 0% | **62.50**% | 50.70% | **89.30%** |
| **Path Time Ratio** | random | 100% | 123.61% | **107.65**% | 123.61% | **107.65%** |
| | hard | 100% | / | **128.22**% | 276.25% | **128.22%** |
| **Collision Checks** | random | 60K | 55.88 | **17.17** | 55.88 | **17.17** |
| | hard | 1081K | / | **52.68** | 1161.29 | **52.68** |

We also provide the success rate of **GNN-TE** with backtracking in all the environments in 7. As the DoF and the complexity of the configuration space increase, the searching space grows and requires more backtracking steps. Thus the increase in success rate by backtracking may not be as significant as in the simple settings if we keep the backtracking steps the same. However, GNN-TE still shows a significant advantage over Dijkstra-H even with backtracking in all the settings.

Table 7: Success rates of GNN-TE and Dijkstra-H with backtracking search

| | | 2Arms | Kuka-4DoF | Kuka-5DoF | Kuka-7DoF | 3Arms | Kuka3Arms |
|---|---|---|---|---|---|---|---|
| **random** | **Dijkstra-H** | 89.7±0.03 | 96.3±0.01 | 96.2±0.01 | 97.7±0.01 | 85.9±0.01 | 93.9±0.01 |
| | **GNN-TE** | **94.1±0.02** | **97.8±0.01** | **97.6±0.00** | **98.8±0.01** | **92.1±0.01** | **97.4±0.01** |
| | **Dijkstra-H w. BT** | 94.1±0.01 | 96.7±0.02 | 96.2±0.01 | 97.8±0.01 | 92.4±0.01 | 94.2±0.01 |
| | **GNN-TE w. BT** | **98.0±0.01** | **97.8±0.01** | **97.7±0.00** | **98.9±0.01** | **97.1±0.01** | **97.4±0.00** |
| **hard** | **Dijkstra-H** | 0.0±0.0 | 0.0±0.0 | 0.0±0.0 | 0.0±0.0 | 0.0±0.0 | 0.0±0.0 |
| | **GNN-TE** | **62.5±0.02** | **34.9±0.00** | **37.9±0.15** | **38.1±0.12** | **52.5±0.03** | **42.1±0.08** |
| | **Dijkstra-H w. BT** | 50.7±0.06 | 10.1±0.01 | 5.8±0.40 | 2.8±0.24 | 45.8±0.05 | 2.7±0.01 |
| | **GNN-TE w. BT** | **89.3±0.03** | **36.4±0.00** | **40.8±0.15** | **39.4±0.12** | **82.6±0.02** | **44.3±0.01** |

## C.4 Comparison with End-to-End RL

We compare our approach with RL-based approaches, **DQN-GNN** and **PPO-GNN** specifically. The two algorithms both encode the graph using GNN as ours in stage 1. **DQN-GNN**, similar to our

local planner, learns a network to evaluate the Q value of the subsequent edge as a priority value. **PPO-GNN** learns a policy network that output the next configuration, and we project it onto the nearest vertex on the graph encoded by GNN. We define the reward as $-10$ for collision, $10$ for reaching the goal, and the $distance\ displacement$ for non-collision configurations.

In the general RL setting, we do not expect the generalization capability of algorithms. But as a general graph encoder, GNN can achieve generalization between graphs. Based on this, we train **DQN-GNN** and **PPO-GNN** across problems and test their performance. On training set, **DQN-GNN** achieves $54.5\%$ success rate while **PPO-GNN** only achieves $21.1\%$. We also provide results on randomly generated test cases and hard cases in Table 8. We can observe that **GNN-TE** significantly outperforms all the RL approaches. Moreover, the advanced inductive bias of GNN for discrete decision-making problems explained why **DQN-GNN** has better performance than **PPO-GNN** in both randomly sampled cases and hard cases. Nevertheless, **DQN-GNN** and **PPO-GNN** both cannot efficiently find plans, especially in hard cases. This is because RL-based approaches have trouble finding a feasible path without demonstration from the oracle and only rely on rewards in challenging problems.

Table 8: Table for RL Approaches in 2Arms Environment

| | | GNN-TE | DQN-GNN | PPO-GNN |
|---|---|---|---|---|
| **Success Rate** | random | **94.10%** | 62.40% | 9.80% |
| | hard | **62.50%** | 2.00% | 0.70% |
| **Path Time Ratio** | random | **103.55%** | 105.47% | 119.73% |
| | hard | **102.43%** | 109.76% | 134.15% |
| **Collision Checking** | random | **4.98** | **4.68** | 5.43 |
| | hard | **6.00** | **6.00** | 7.00 |

### C.5 Comparison with OracleNet-D

We compare **GNN-TE** with a learning-based approach **OracleNet-D** by modifying **OracleNet** [1] to the dynamic version. Concretely, we concatenate the trajectories of obstacles to the input in every roll-out of **OracleNet** to inform the network of the dynamic environment [1]. We provide the results in 2Arms environment in Table 9. (For a fair comparison, we present the result of GNN-TE without DAgger. And the collision checking is not provided because OracleNet-D generates and rolls out the path iteratively without checking the collision.)

Table 9: Table for GNN-TE and OracleNet-D in 2Arms Environment

| | | SIPP | Dijkstra-H | GNN-TE | OracleNet-D |
|---|---|---|---|---|---|
| **Success Rate** | random | 100% | 89.70% | **94.10%** | 53.90% |
| | hard | 100% | 0.00% | **58.10%** | 10.80% |
| **Avg Path Time Ratio** | random | 100% | 120.61% | **113.94%** | 1130.76% |
| | hard | 100% | / | **118.92%** | 813.00% |

We observe that the performance of **OracleNet-D** falls behind **GNN-TE** both on success rate and the average time ratio. This result shows that encoding environmental information is important for the planner in a dynamic environment. As mentioned in [1], the configuration of the robot and the environmental information form different distributions and the mapping is challenging. We believe GNN with the attention mechanism and temporal encoding provides a good solution to the problem. Also, GNN-TE benefits from the second-stage local planner, which takes local temporal obstacle information into consideration.

---

[1]We use the original code from repository https://github.com/mayurj747/oraclenet-analysis

## C.6 Ablation Study on Varying Training Set Sizes

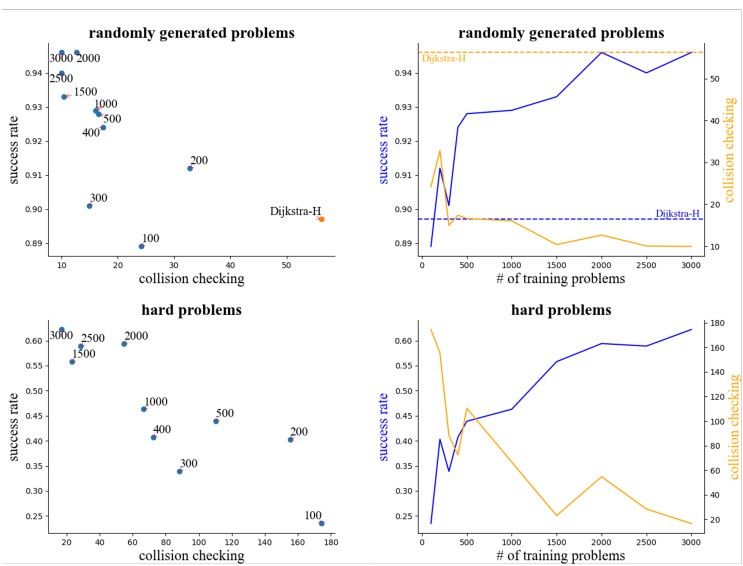

Figure 7: Results on varying training set sizes in 2Arms environment. We observe that **GNN-TE** benefits from increasing the training problems, both in better success rate and less collision checking. **Left:** A scatter plot visualizes the relevance between the success rate and the collision checking regarding the training size. The number on each point indicates the training size. **Right:** An equivalent plot that clearly shows the performance boost benefited from a larger training size. Higher success rate (blue curve) and lower collision checking (orange curve) are favored.

We train **GNN-TE** on varying training problems (specifically 100, 200, 300, 400, 500, 1000, 1500, 2000, 2500, 3000) and test on the same random sampled and hard problems in 2Arms environment.

We observe that **GNN-TE** benefits from increasing the training problems, both in better success rate and less collision checking. From the plot in the right column of the figure, we observed that the trends are prone to be log-like. It shows that the performance will be saturated as the training set covers the problem distribution.

## C.7 Ablation Study on Basic GNN

In Table 10, we provide the overall performance gain by all the components of **GNN-TE** over the basic GNN (**GNN-basic**) in 2Arms environment. Specifically, in the first stage, **GNN-basic** removes the attention mechanism and temporal encoding. And in the second stage, **GNN-basic** only inputs the obstacle encoding at the current time step.

Table 10: Overall performance gain over basic GNN

|  |  | SIPP | Dijkstra-H | GNN-TE | GNN-basic |
|---|---|---|---|---|---|
| **Success Rate** | random | 100% | 89.70% | **94.10%** | 92.70% |
|  | hard | 100% | 0.00% | **62.50%** | 32.00% |
| **Avg Path Time Ratio** | random | 100% | 123.73% | **107.78%** | 112.42% |
|  | hard | 100% | / | **122.13%** | 185.92% |
| **Avg Collision Checking** | random | 60K | 56.21 | **17.44** | 28.80 |
|  | hard | 1081K | / | **45.23** | 109.70 |

## C.8 Failure Modes in 2Arms Environment

We provide visualizations of **GNN-TE** failing to find feasible solutions in 2Arms environments. We find there are mainly two modes: it fails to make a detour in Fig. 8 or gets too close to the moving obstacles in Fig. 9. In Fig. 8, we can observe that **GNN-TE** plans to directly get to the goal while the feasible path is to make a detour to avoid the obstacle. In Fig. 9, although **GNN-TE** can follow the correct direction but fail in getting too close to the obstacle arm.

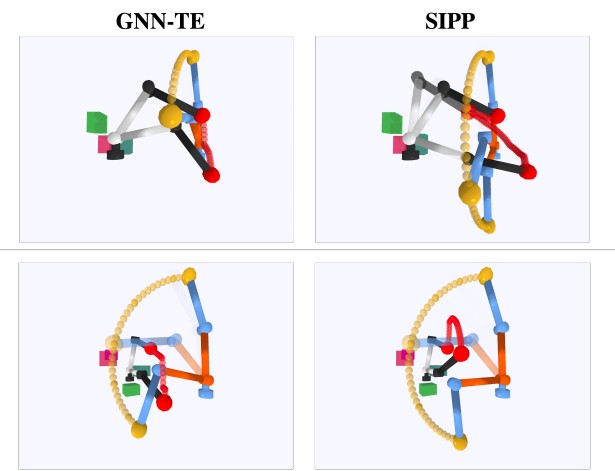

Figure 8: Failure mode: the planner fails to make a detour. Our planner controls the arm in black and white.

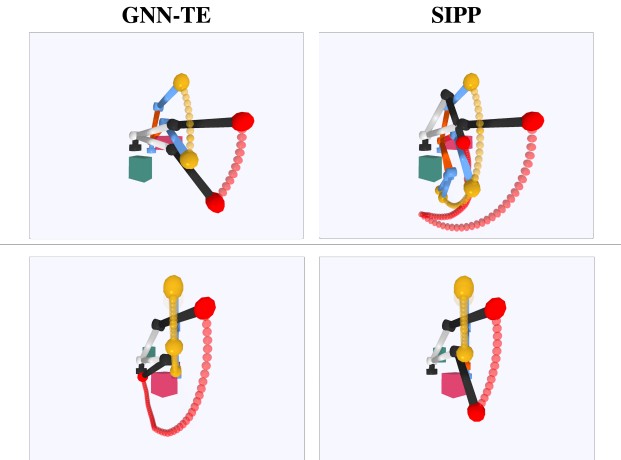

Figure 9: Failure mode: the planner gets too close to the obstacle. Our planner controls the arm in black and white. Though the planner follows the correct direction, it gets too close to the obstacle arm, which leads to the collision.

# Appendix D    Limitations and Future Work

## D.1 Discussions on Using GNNs and Attention Mechanism

Motion planning has been a longstanding challenge in robotics, especially in dynamic environments. Our approach uses a learning-based approach leveraging Graph Neural Networks to efficiently tackle this problem. GNNs show great capability in capturing geometric information and are invariant to the

permutations of the sampled graph. Another challenge in dynamic environment is that the difference in distributions of the robot configuration and the environmental information makes the mapping and motion planning challenging. Our approach tackles this by introducing the attention mechanism with temporal encoding to learn the correlation between the temporal positions of obstacles and the ego-arm configuration on the graph. It is efficient because, as for a configuration node on the graph, the obstacles' positions of some time steps are more important than others, as the obstacles may have more possibilities of colliding with the ego-arm at those time steps. So, in this case, the obstacles of those time steps should be given more importance in modeling. Also, the attention mechanism can take time sequences with variable length as inputs.

Regardless of the empirical efficiency, the performance of the GNN-based approach is still bounded by the sampled configurations. It can only be boosted by a sufficient number of nodes on the graph, especially in a complex environment and with a robot with a high degree of freedom. It is still an open problem how many samples would be sufficient for the GNN to capture the geometric pattern from the configuration space.

## D.2 Limited Performances on Hard Problems

Although **GNN-TE** can achieve a better success rate than other learning-based approaches, it is still not complete and has limited performance on hard problems (see Section C.8 for examples of failure modes). A direction of solving this problem is to do hard example mining and train on those problems, where we train **GNN-TE** on extra hard examples and test its performance, and the success rate rises from 62.5% to 71.3% on 2Arms environment. However, in general, we believe the safety and reliability of learning-enabled systems are always a core issue that needs to be solved after learning-based approaches show clear benefits.

For motion planning, a potential future direction is to integrate our learning-based component with monitoring. Such monitoring identifies hard graph structures that are out-of-distribution for the neural network components. It ensures that the learning-based components are only used when the planning can be safely accelerated, in which case they will provide great benefits in reducing collision checking and overall computation. When hard or out-of-distribution cases occur, the planner should fall back to more complete algorithms such as **SIPP**. There also has been much ongoing development in frameworks for ensuring the safe use of learning-based components in planning and control, which we believe is orthogonal to our current work. For example, [3] provides reviews learning-based control and RL approaches that prioritize the safety of the robot's behavior.

## D.3 Trade-off Between Quality and Efficiency

Another observation from the result is the trade-off between quality (success rate of finding paths) and efficiency (number of collision checks). In this work, we further add backtracking, where we keep a stack of policy edges of the top-n priority values and allow the algorithm to take the sub-optimal choices if it fails. Therefore, the backtracking will increase the collision checking with the hope of finding a solution. Although adding this or other systematic searching algorithms can improve the quality in the sacrifice of efficiency, we think the actual bottleneck might still be the priority values as the heuristic produced by the model. We believe this trade-off may be a crucial learning-based dynamic motion planning topic and needs future investigations.

## D.4 Problem Distribution and Generalization

As most learning-based approaches would assume, our model needs to be trained on the same actor and obstacle arms as it's tested on. Both the sampled graph and the expert trajectory are implicitly conditioned on the kinematic structure. This assumption aligns with the most immediate use of learning-based components for reducing repeated planning computation in a relatively fixed setting of arm configurations. We believe learning planning models that can be generalized to arbitrary arms and obstacles is still challenging for the community, for it requires an in-depth study of other issues that have not been fully understood, such as the inherent generalization properties of graph neural networks. As shown in [9], there still exists the trade-off between expressivity and generalization in GNN. We leave this topic to future works.

# Appendix E    More Snapshots in Different Environments

In this section, we show more snapshots of baselines SIPP, GNN-TE, Dijkstra-H in different environments. In those environments and cases, **GNN-TE** succeeds in finding a near-optimal path while **Dijkstra-H** fails.

**SIPP**          **GNN-TE**          **Dijkstra-H**

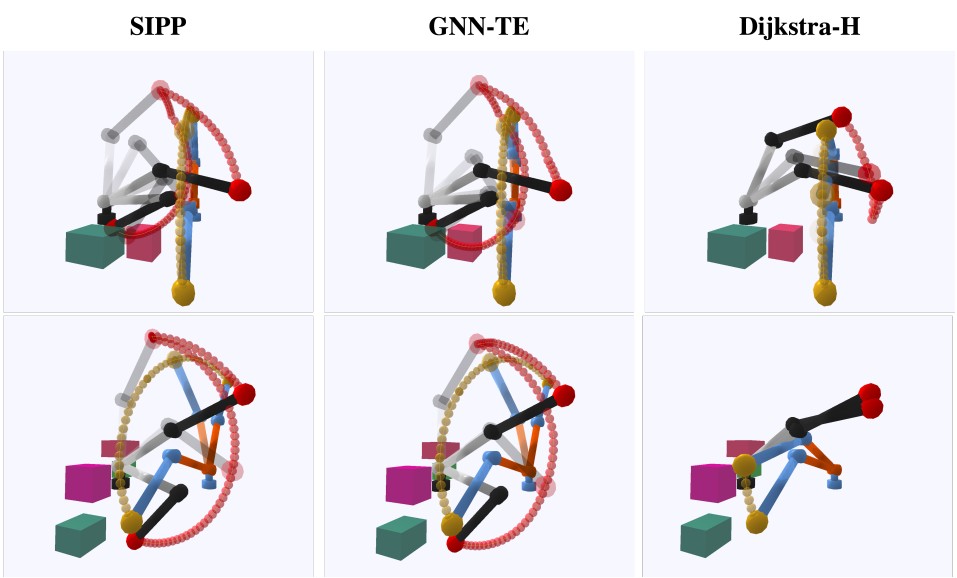

Figure 10: Snapshots: 2Arms

**SIPP**          **GNN-TE**          **Dijkstra-H**

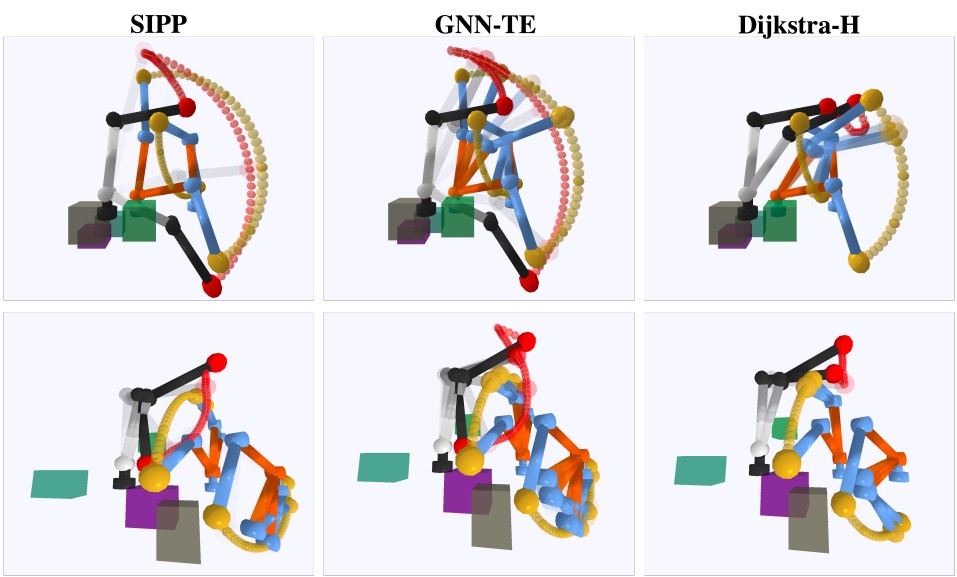

Figure 11: Snapshots: 3Arms

| SIPP | GNN-TE | Dijkstra-H |
|:---:|:---:|:---:|

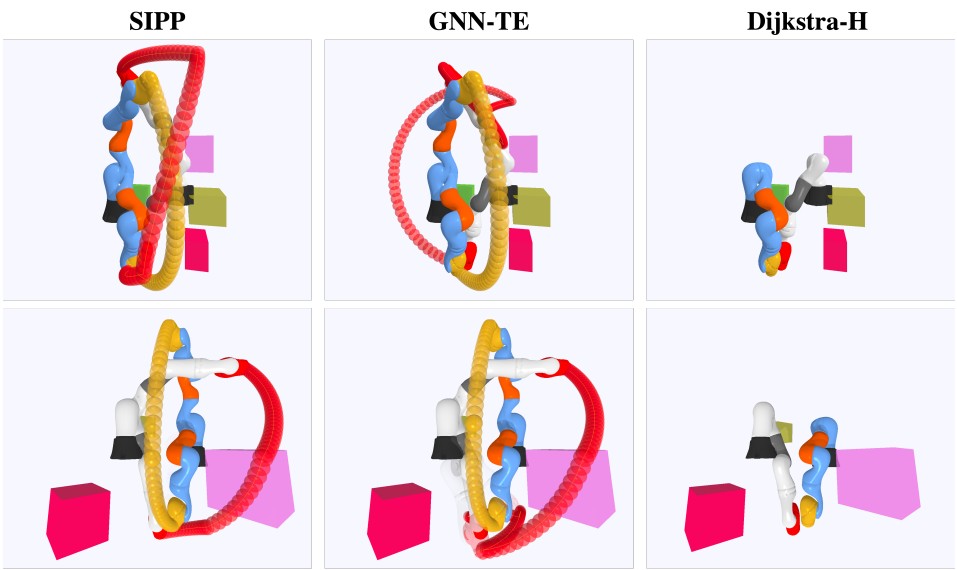

Figure 12: Snapshots: Kuka-4DoF

| SIPP | GNN-TE | Dijkstra-H |
|:---:|:---:|:---:|

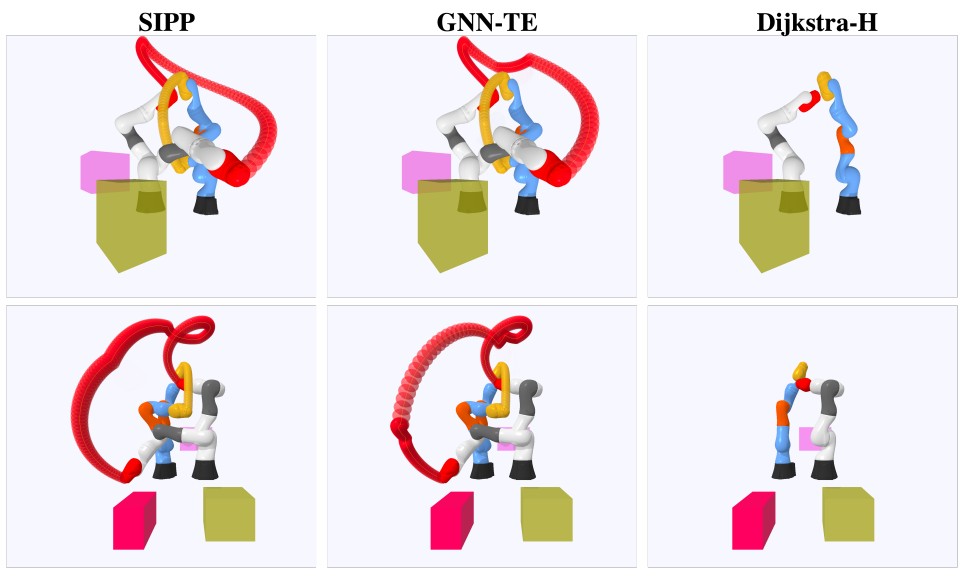

Figure 13: Snapshots: Kuka-5DoF

|  **SIPP**  |  **GNN-TE**  |  **Dijkstra-H**  |

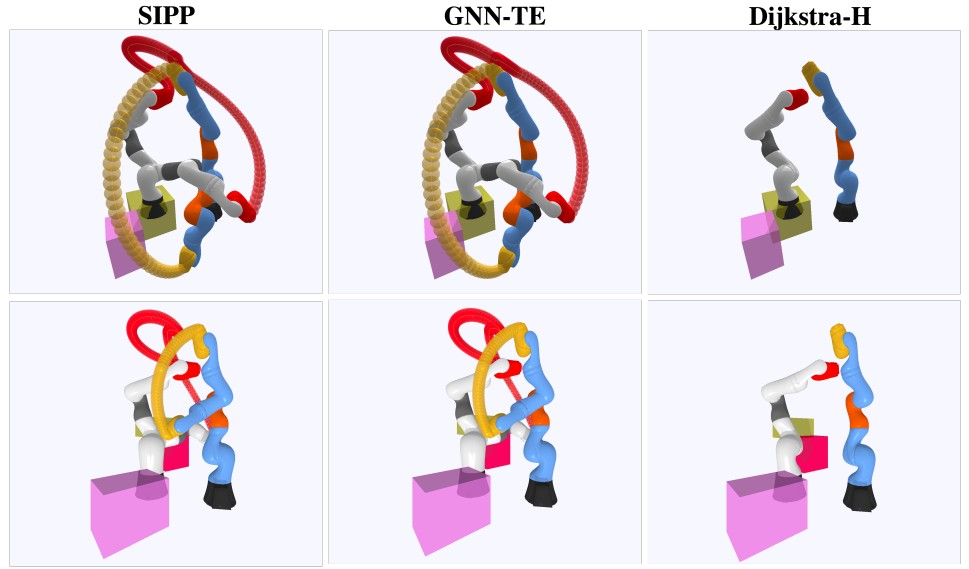

Figure 14: Snapshots: Kuka-7DoF

|  **SIPP**  |  **GNN-TE**  |  **Dijkstra-H**  |

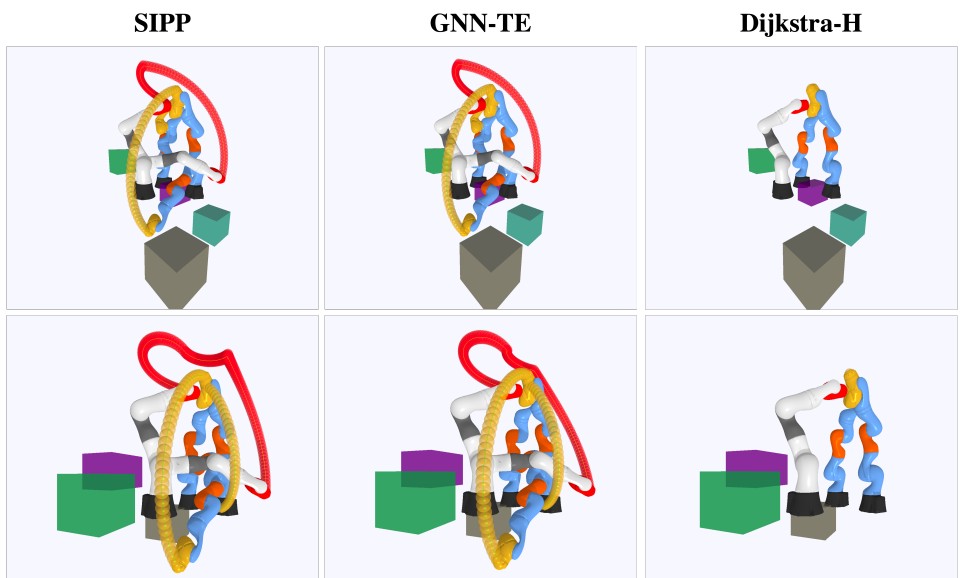

Figure 15: Snapshots: Kuka3Arms