# OpenReview forum: "Learning-based Motion Planning in Dynamic Environments Using GNNs and Temporal Encoding"
_NeurIPS.cc/2022/Conference — NeurIPS 2022 Accept_

### Official Review · Reviewer_oTjD · 2022-07-12

**Rating:** 6
**Confidence:** 4
**Soundness:** 3 good
**Presentation:** 3 good
**Contribution:** 3 good

**Summary:**

This paper proposes an imitation learning approach which mimics the behavior of a search-based configuration space motion planner with dynamic obstacles (SIPP [32]) using DAgger. Planning is performed on a sampled road map graph which is encoded with the motion of a dynamic obstacle and the state of static obstacles in a GNN. The method seems to assume a fixed shape and kinematic structure of dynamic (robot arms) and static (boxes) obstacles. The method is evaluated in simulation and compared in planning performance metrics with ablations, SIPP and a simpler Dijkstra-based baseline. The approach well improves the required computation in terms of collision checks, while keeping a high level of success rate and path efficiency.

**Questions:**

* l. 226 it's unclear how the Dijksta-H baseline works from the description. How does it handle arrival time and does it use collision checking ? Please revise and provide more implementation details in the supplementary material.
* l. 246 is SIPP guaranteed to find optimal paths and for which planning problem ?
* The ablation study in 5.2 only reveals little improvements by the individual components relative to the performance of the baselines. What is the overall performance gain by all the components over the basic GNN ?




**Limitations:**

* The paper does not address limitations and assumptions properly. For instance, it should be clearly stated that the model needs to be trained for the specific actor arm, the obstacle arms and the static obstacles which it is tested on, i.e. that specific kinematic structures and obstacle shapes are assumed. How can the approach be generalized to arbitrary obstacle shapes and arm kinematics?

**Strengths And Weaknesses:**

Strengths:
* The approach seems novel and effective in cloning the planner behavior for the trained environment settings.
* The paper is well structured and can be easily followed.
* Improvements over ablations, the search-based planning baseline SIPP, and a simpler Dijkstra based planner are demonstrated in several planning problems with various robot kinematics and difficulties.

Weaknesses:
* The writing requires significant proof reading to correct spelling and grammar mistakes.
* The title mentions "Manipulation Planning" but actually there is no object manipulation performed. The approach performs motion planning. Title should be changed accordingly.
* The supplementary material should provide the network details to allow for reproducibility of the work. Ideally, also code and datasets should be provided publicly.
* l.115 the explanation of where the attention mechanism will be used is confusing, because the concepts are unclear at that point of reading. Just introduce the attention mechansim abstractly.
* l. 143, please introduce the subscript notation v_i,v_j, x_i, x_j and that the subscript links nodes and respective components of the features. the y needs to be separated with a comma.
* The notation O = O + TE is unclear. first of all, this is not a proper equation but an assignment. Rather use a new symbol for the left O. Also, O and TE as well as the + operation are not defined.

---

> ### Author Response · Authors · 2022-08-01
> **Response to Reviewer oTjD**
>
> Thank you for your careful review and acknowledgment of novelty and effectiveness. We address the specific questions as follows.
>
> > The writing requires significant proof reading to correct spelling and grammar mistakes.
>
> Thank you for pointing them out. We have updated the paper with more thorough proofreading and will continue to improve the writing.
>
> > The title mentions "Manipulation Planning" but actually there is no object manipulation performed...Title should be changed accordingly.
>
> Thanks for pointing this out. We were trying to emphasize the context of high-dimensional planning problems with robot arms, which is different from the simpler problem of navigating in 2D environments. We have changed the title in the updated paper.
>
> > The supplementary material should provide the network details to allow for reproducibility of the work. Ideally, also code and datasets should be provided publicly.
>
> We have provided the network details in Appendix B. The code and data will be fully made open-source upon publication of the work.
>
> > l.115 the explanation of where the attention mechanism will be used is confusing...
> l. 143, please introduce the subscript notation v_i,v_j, x_i, x_j ...
> The notation O = O + TE is unclear. ...
>
> We have organized and rephrased the mentioned parts in the updated version.
>
> > l. 226 it's unclear how the Dijksta-H baseline works from the description. How does it handle arrival time and does it use collision checking ? Please revise and provide more implementation details in the supplementary material.
>
> We have provided the pseudo-code of Dijkstra-H in the updated Appendix A. A brief description is as follows. Similar to our GNN-TE, Dijkstra-H first samples multiple configurations of the ego-arm to form a graph along with the start and goal node. It uses the Dijkstra algorithm to calculate the shortest distance from every node to the goal. When planning, the ego-arm will follow this heuristic and prioritize the node with the shortest distance to the goal as the next step, while checking collision on edge and keeping track of the time. If there is a collision when moving to the target node, Dijkstra-H will turn to query the next closest node, and so on. Dijkstra-H fails when it cannot find any next available nodes.
>
>
> > l. 246 is SIPP guaranteed to find optimal paths and for which planning problem ?
>
> Yes, SIPP is guaranteed to find optimal paths, for planning problems in dynamic environments with known obstacle trajectories. We refer readers to Section III.B of [32] for more details.
>
>
>
>
>
>
>
>
>
>
>
> > The ablation study in 5.2 only reveals little improvements by the individual components relative to the performance of the baselines. What is the overall performance gain by all the components over the basic GNN ?
>
> We provide all components' overall performance gain over basic GNN (GNN-basic) here and in the updated Appendix C.7. Specifically, in the first stage, the basic GNN has no attention mechanism and temporal encoding. And in the second stage, it only inputs the obstacle encoding at the time step.
>
> |                     |        | SIPP  | Dijkstra-H | GNN-TE | GNN-basic |
> | :---: | :---: |---: |---: |---: |---: |
> | **Success Rate**       | random | 100% | 89.70% | **94.10%**  | 92.70% |
> |                     | hard   |100%|  0%   | **62.50%** | 32.00% |
> | **Avg Path Time Ratio** | random | 100%|123.73%|**107.78%**|112.42%|
> |                     | hard   |  100%|/|**122.13%**|185.92%|
> | **Avg Collision Checking** | random | 60K|56.21|**17.44**|28.80|
> |                     | hard   |1081K|/|**45.23**|109.70|
>
>
> > The paper does not address limitations and assumptions properly. For instance, it should be clearly stated that the model needs to be trained for the specific actor arm, the obstacle arms and the static obstacles which it is tested on, i.e. that specific kinematic structures and obstacle shapes are assumed. How can the approach be generalized to arbitrary obstacle shapes and arm kinematics?
>
> Thanks for pointing this out. Since our method is learning-based, it should be trained on the same actor and obstacle arms as it’s tested on, as both the sampled graph and the expert trajectory are implicitly conditioned on the kinematic structure. This assumption is reasonable as the most immediate use of learning-based components is for reducing repeated planning computation in a relatively fixed setting of arm configurations. We believe learning planning models that can be generalized to arbitrary arms and obstacles may be too ambitious for now, and it requires an in-depth study of many other issues that have not been fully understood, such as the inherent generalization properties of graph neural networks. We have provided more discussions in Limitations and Future Work in the updated Appendix D.

---

### Official Review · Reviewer_sJvQ · 2022-07-13

**Rating:** 6
**Confidence:** 4
**Soundness:** 2 fair
**Presentation:** 3 good
**Contribution:** 3 good

**Summary:**

This paper proposes to use a learning-based approach for motion planning with dynamic obstacles. Each possible robot configuration (after discretization) is represented as a node. This method assumes it knows the full trajectory of the obstacles. It first generate expert trajectories using some sampling-based path planing method (SIPP) and use that as the ground-truth and try to imitation that policy. The performance is further enhanced by using DAGGER. The advantage of using learning-based approach to imitation the tradition approach is that the learning-based approach is much faster.

**Questions:**

The following questions are essentially what I said in the above section:

1. How is the obstacle trajectory represented?

2. How to deal with cases where the obstacles' trajectories are unknown?

3. Is there any possible comparison with other learning-based methods?

**Strengths And Weaknesses:**

[originality] This work is inspired by the transformer literature and successfully modify it to the motion planning community. I believe this paper has original contribution for this adapatation.

[quality] I believe overall this paper gives detailed and comprehensive evaluation of the proposed method. However, I found the comparison between the proposed method and other learning-based method is missing. For example, one simple baseline I could come up is using [1]’s network, with an additional input of all the possible obstacle trajectory (since the full trajectory is known), and output the next optimal configuration.

[clairty] I think the technical part is clear except for one point: this paper claims previous method considers fixed graphs ([39, 21]). Then I’m wondering how the obstacle trajectory is represented? My understanding is the obstacle’s full trajectory is represented as a node in the graph. However, in that case the graph is fixed and some previous methods need to be considered in the experiments (such as [21, 39]). The authors may want to clarify this.

[significance] The proposed method is much master than the traditional optimization-based method. This can be a great advantage for many applications. However, the performance drop seems to be huge in the hard case (Figure 4 and Table B2). Since motion planning would be safety-critical in many cases. This may be a major limitation for the proposed method. I think it would be better to provide some illustrations and convincing improvement direction in the paper. Another possiblity to make the contribution more significant is to consider cases where the object trajectories are not known thus prediction and rapid re-planning is needed.

---

> ### Author Response · Authors · 2022-08-01
> **Response to Reviewer sJvQ**
>
> Thanks for your careful review and important suggestions for the comparison with learning-based approaches. We address the specific questions as follows.
>
> > How is the obstacle trajectory represented? … The authors may want to clarify this.
>
> The dynamic obstacle trajectory is represented by a vector of all the joint positions in the workspace in a time window (as mentioned in l.106-107). We only maintain a graph of all the sampled configurations of the ego-arm, while the trajectory is not represented as a node on the graph. Instead, we introduce the trajectory of obstacles, which can be of arbitrary length, using the attention mechanism to encode the information into the nodes and edges of the configuration graph (as illustrated in Figure 2).
>
> The approaches in [21, 39] directly use fixed graph structures to update the internal values. Since no temporal information is introduced in their architecture, the output of the network is invariant to different time steps, which might not apply to the dynamic environment, since some edges can be feasible to traverse at some time steps but infeasible at the other time steps.
>
> Our architecture does not directly generate the outputs solely based on the fixed graph, and provides a temporally-sensitive solution by introducing the second-staged planner. It uses local-level information for the network to plan towards the goal vertex while also considering the dynamic obstacles to avoid collisions. We believe this is one of the key components of the dynamic environment.
>
> > However, I found the comparison between the proposed method and other learning-based method is missing...
>
> We provided comparisons with RL baselines in Section 5.1 in the paper and C.4 in the updated Appendix. In general, we found that GNN-based approaches outperform learning-based approaches that do not exploit the graph structure. For instance, we have further compared with another baseline OracleNet in the dynamic environment (OracleNet-D), which is modified from [1], with the trajectories of obstacles as additional inputs in every rollout of the network. We use the original repository of the paper and provide the result and analysis in C.5 in the updated Appendix.
>
> > However, the performance drop seems to be huge in the hard case (Figure 4 and Table B2)... I think it would be better to provide some illustrations and convincing improvement direction in the paper.
>
> We have visualized some failure cases in 2Arms environment (See C.8 in the updated Appendix). We find there are mainly two failure modes: it fails to make a detour in Fig.8 or gets too close to the moving obstacles in Fig.9. In Fig. 8, we can observe that GNN-TE plans to directly get to the goal while the feasible path is to make a detour to avoid the obstacle. In Fig.9, although GNN-TE moves in the correct direction but fails in getting too close to the obstacle arm.
>
> We have provided more discussions on improvement direction in the updated Appendix D. We put the same discussions here as follows. Because the trained policy using the learning-based method relies on the training distribution, and in the paper, we train on the randomly generated cases and test on both random and hard ones. Therefore, the performance on the hard examples may not be as prominent as the random ones. We’ve tried to train the algorithm on extra hard examples and test its performance, and the success rate rises from 62.5% to 71.3% on 2Arms environment.
>
> In general, we believe the safety and reliability of learning-enabled systems are always a core issue that needs to be solved after learning-based approaches show clear benefits. For motion planning, a potential future direction is to integrate our learning-based component with monitoring. Such monitoring identifies hard graph structures that are out-of-distribution for the neural network components. It ensures that the learning-based components are only used when the planning can be safely accelerated, in which case they will provide great benefits in reducing collision checking and overall computation. When hard or out-of-distribution cases occur, the planner should fall back to more complete algorithms such as SIPP. There has been much ongoing development in frameworks for ensuring the safe use of learning-based components in planning and control, which we believe is orthogonal to our current work.
>
> > Another possibility to make the contribution more significant is to consider cases where the object trajectories are not known thus prediction and rapid re-planning is needed.
>
> This would be a great next step for our work, which would bring us out of the context that SIPP is typically used for. We can add pre-trained prediction modules using Gaussian Processes or deep neural models, and use the output of it as the pseudo obstacle input of GNN-TE. In practice, our GNN component only requires around 0.1~0.2 seconds to infer for the test case on average, and has the potential to be used for rapid re-planning.

---

> > ### Comment · Reviewer_sJvQ · 2022-08-04
> > **Re: Paper 3708 Authors**
> >
> > Thanks for your response. My concerns are well-addressed.
> >
> > Previously I was confused about the definition of ‘fixed graph’ (line 91) because:
> > 1. in the Stage 1, the graph is irrelevant to timestep t.
> > 2. the connection between nodes does not change at all.
> > I now understand the it means in the second state, the feature is time-dependent so ‘not fixed’.
> > However, I suggest the author clarify this point in the text because when the reader reads ‘learning to explore edges given fixed graphs [41, 23]’, they may think your work operates on a dynamic graph where the graph connectivity changes.
> >
> > I think including comparison with OrcleNet makes the paper more complete and solid.
> >
> > Regarding the demonstration of failure case, I would highly encourage you to have some videos instead of image snapshots in the final version. Currently it is a bit hard to imagine the whole trajectory especially because there are dynamic obstacles.

---

> > > ### Author Response · Authors · 2022-08-04
> > > **Response to Reviewer sJvQ**
> > >
> > > We thank the reviewer for the important questions on the definition and baseline, and for reading our response. We will clarify the definition and add videos in the final version.

---

### Official Review · Reviewer_4Jb4 · 2022-07-23

**Rating:** 6
**Confidence:** 3
**Soundness:** 3 good
**Presentation:** 3 good
**Contribution:** 2 fair

**Summary:**

The paper aims to improve the efficiency of motion planning problems in dynamic environments, such as with moving obstacles or multiple manipulators. The approach taken is graph neural networks and temporal encoding framework. Trained with imitation learning and data aggregation procedures, the proposed methods significantly reduce collision checks and planning time, at the cost of reducing planning success rate.

Concretely, the core building blocks are graph neural networks, attention mechanism, temporal encoding of moving obstacles, and imitation learning with DAGGER style of data aggregation. Extensive experiments are provided to compare the proposed method with search-based baselines, ablation baselines, and end-to-end RL methods.

**Questions:**

1. How is the speed (saving number of collision checks) in trade-off with quality (success rate of finding paths), and how does this trade-off vary with different amount of training? E.g., currently 2000 problems are used for training. Is the performance increasing with more training problems? Is it in a linear or log trend? Does it saturate?

2. The separation between random and hard cases is determined by Dijkstra-H. Is there a justification of this choice or because there are no better ways? It's also helpful to provide a direct measure on the complexity of the planning problems.

3. Using the attention mechanism to encode obstacles is not analyzed separately, i.e., why is attention mechanism the most proper to model the obstacles?

**Limitations:**

No specific analysis or discussion is provided on the limitations by the authors. Though, from the result section, it can be observed the proposed GNN based motion planning has a reduced success rate compared to the baseline SIPP. More comprehensive analysis on this would be greatly appreciated by the community. Refer to Questions.

**Strengths And Weaknesses:**

Originality

This paper takes a graph neural network (GNN) approach and temporal encoding to tackle motion planning in dynamic environments. It is a novel way of applying existing approaches such as GNN and attention mechanisms to an existing problem. Many details were reasonably designed, e.g., the temporal encoding of moving obstacles.

Quality and Clarity

The algorithm is reasonably designed, justified by the provided ablation studies of module and data flow choices. The approach is clearly presented, backed with diagrams and example drawings.

There are quite a few typos throughout the paper. Please do more proofread for future revisions. Just to name a few, "(" in L112, why (k) is needed for the aggregation function in eq(1)? L118 "has to". 165 "E_j" -> "E_i".

Significance

The problems of motion planning in dynamic environments is an important problem, relevant for a wide group in the community. Leveraging GNNs (likely the first time) to solve this problem is interesting. If the pros and cons are thoroughly analyzed and presented, it can be of greater significance.

This is an important issue to address. It is not a surprise using a GNN and temporal embedding, trained on the same set of static and dynamic objects, can achieve with more efficient search heuristics. However, this efficiency benefits comes at a price, sacrificing the success rate, i.e., paths cannot be found in many cases. The authors did an attempt using backtracing, but only showed the result on the most simple setting, two-DOF arms. Yet it is not strong result (98% for random and 89% for hard). From a reader's point of view, understanding this trade-off in a more systematic way would be a key contribution. See also Questions.

---

> ### Author Response · Authors · 2022-08-01
> **Response to Reviewer 4Jb4**
>
> Thank you for your careful reading. We are glad you found our GNN-based method to be a novel solution to an important problem. We address the specific questions as follows.
>
> > There are quite a few typos throughout the paper. Please do more proofread for future revisions.
>
> Thanks for pointing them out. We have done more proofreading and fixed the typos in the updated version.
>
> > If the pros and cons are thoroughly analyzed and presented, it can be of greater significance.
>
> We have included further discussions of the pros/cons and limitations in the updated Appendix D.
>
> > The authors did an attempt using backtracing, but only showed the result on the most simple setting, two-DOF arms.
>
> We refer the reviewer to Section C.3 (Table 7) in the updated Appendix. As the DoF and the complexity of the configuration space increase, the searching space grows and requires more backtracking steps. Thus the increase in success rate by backtracking may not be as significant as in the simple settings if we keep the backtracking steps the same. However, GNN-TE still shows a significant advantage over Dijkstra-H even with backtracking in all the settings.
>
> > How is the speed (saving number of collision checks) in trade-off with quality (success rate of finding paths), and how does this trade-off vary with different amount of training? E.g., currently 2000 problems are used for training. Is the performance increasing with more training problems? Is it in a linear or log trend? Does it saturate?
>
> Thank you for mentioning the trade-off between the number of collision checking and the success rate. We have included more experimental results in the updated Appendix C.6, showing that the success rate increases and the collision rate decreases as the training size increases, both approximately in logarithmic trend. A brief description of this additional experiment is as follows. We train GNN-TE on varying training problems (100, 200, 300, 400, 500, 1000, 1500, 2000, 2500, 3000) and test on the same random sampled and hard problems in 2Arms environment. We observe that GNN-TE benefits from increasing the training problems, both in better success rate and less collision checking. We also provide the trends of the two criteria in the right column of the figure, and we believe both the trends are prone to be log-like. It shows the performance will be saturated as the training set covers the problem distribution.
>
> During inference, GNN-TE acts in a greedy way to follow the edges with the highest priority value. By backtracking, we keep a stack of policy edges of the top-n priority values and allow the algorithm to take the suboptimal choices if it fails. Therefore, the backtracking will increase the collision checking with the hope of finding a solution. Although adding this or other systematic searching algorithms boosts the quality in the sacrifice of speed, we think the actual bottleneck might still be the priority value as the heuristic produced by the model. We believe this trade-off may be a crucial learning-based dynamic motion planning topic and needs future investigations.
>
> > The separation between random and hard cases is determined by Dijkstra-H… It's also helpful to provide a direct measure on the complexity of the planning problems.
>
> Intuitively, an instance is hard if using greedy actions fails, i.e., when the shortest path to the goal needs to be avoided to find feasible paths. The typical hard cases considered in motion planning, such as U-shaped barriers, all share this characterization. Right now, we are using Dijkstra-H as an empirical metric to reflect this consideration. A fully accurate formal measure may not be easily definable because to do so, we need to basically separate the classes of simple planning problems (say polynomial-time solvable instances, i.e., in complexity class P) from the general dynamic motion planning problem (which is PSPACE-complete [35]), which is related to separating P from PSPACE.
>
>  > Using the attention mechanism to encode obstacles is not analyzed separately, i.e., why is attention mechanism the most proper to model the obstacles?
>
> Since the dynamic obstacles form a trajectory in the time dimension, attention mechanisms make it possible to learn the correlation between the position of obstacles at every time and the ego-arm configuration on the graph. This is an empirical observation, and we have included further discussions in the updated Appendix D.1.
>
> > No specific analysis or discussion is provided on the limitations by the authors. …More comprehensive analysis on this would be greatly appreciated by the community.
>
> We have provided the limitations and future work in the updated Appendix D.

---

> > ### Comment · Reviewer_4Jb4 · 2022-08-04
> > **Re: Paper 3708 Authors**
> >
> > Thanks for the response! My questions are addressed.
> >
> > It's interesting seeing the trade-off "curve" with different number of training examples. The added experiments also suggested the inadequacy of backtracing in harder environments.

---

> > > ### Author Response · Authors · 2022-08-04
> > > **Response to Reviewer 4Jb4**
> > >
> > > We thank the reviewer for the crucial questions and carefully reading our response! As also discussed in Appendix D, we think the topics on planning in more challenging environments and problems will be a great direction for our future work.

---

### Meta-Review · Area_Chair_edi7 · 2022-09-01

**Recommendation:** Accept
**Confidence:** Certain

**Metareview:**

Robot motion planing in dynamic environments remains a significant problem. All reviewers consistently agree that the suggest GNN approach in this paper has useful merits, is of general interest, and that the paper is above the publication threshold. Detailed comments of the reviewers provide a good source for some fine-tuning improvements of the paper.

**Award:**

No

---

### Decision · Program_Chairs · 2022-09-14

Accept